# Unleashing the Power of Large Language Models for Denoising Recommendation

## ABSTRACT

Recommender systems are vital for personalizing user experiences, yet they often rely on implicit feedback data that can be noisy and misleading. Existing denoising studies typically involve either incorporating auxiliary information or learning denoising strategies from interaction data. Nonetheless, they face challenges due to the inherent limitations of external knowledge and interaction data, as well as the non-universality of certain predefined assumptions, which hinder their ability to accurately identify noise. Recently, large language models (LLMs) have garnered significant attention due to their extensive world knowledge and powerful reasoning capabilities. Despite this, the potential of LLMs to enhance the denoising process in recommendations remains largely unexplored. In this paper, we introduce LLaRD, a novel framework that leverages LLMs to improve the denoising process in recommender systems, thereby enhancing overall recommendation performance. Specifically, LLaRD generates denoising-related knowledge by first enriching semantic insights from observational data through LLMs, facilitating a comprehensive inference of user-item preference knowledge. It then employs a novel Chain-of-Thought (CoT) technique over user-item interaction graphs to uncover relation knowledge pertinent to denoising. Finally, it utilizes the Information Bottleneck (IB) principle to align the denoising knowledge generated by LLMs with the recommendation targets, effectively filtering out both data noise and irrelevant knowledge produced by the LLMs. Empirical results demonstrate the effectiveness of our proposed framework, showcasing its superior performance in denoising and recommendation accuracy. The code is available at https://anonymous.4open.science/r/LLaRD-5EE5.

## 1 INTRODUCTION

Recommender systems [17, 27, 34, 54] have become essential for mitigating information overload and delivering personalized services. High-quality interaction data that accurately reflect user preferences play a crucial role in enhancing the performance of these recommendation models. In the context of limited explicit feedback [8, 21], implicit feedback (*e.g.*, click, purchase and views) has emerged as a popular alternative due to its abundance and ease of collection [20, 23]. However, implicit feedback data are often noisy and influenced by various incidental factors, which can hinder their ability to accurately represent user preferences [7, 38, 46].

Permission to make digital or hard copies of all or part of this work for personal or classroom use is granted without fee provided that copies are not made or distributed for profit or commercial advantage and that copies bear this notice and the full citation on the first page. Copyrights for components of this work owned by others than the author(s) must be honored. Abstracting with credit is permitted. To copy otherwise, or republish, to post on servers or to redistribute to lists, requires prior specific permission and/or a fee. Request permissions from permissions@acm.org.

*Conference acronym 'XX, June 03–05, 2018, Woodstock, NY*

© 2018 Copyright held by the owner/author(s). Publication rights licensed to ACM.
ACM ISBN 978-1-4503-XXXX-X/18/06
https://doi.org/XXXXXXX.XXXXXXX

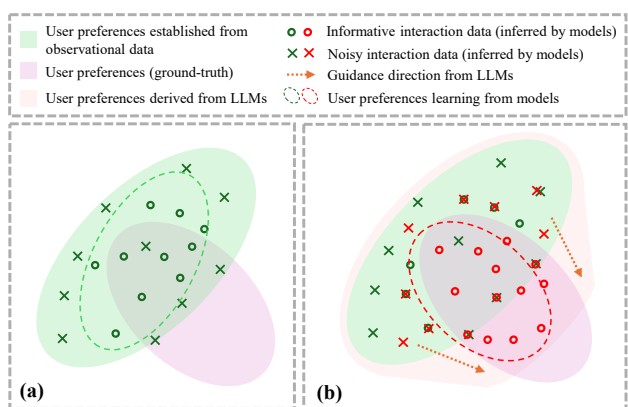

**Figure 1: (a) An intuitive example of learning user preferences from observational data. (b) Improvements of our method (red) over existing methods (dark green).**

For instance, false positive interactions [2, 32] may arise from users' curiosity-driven clicks or unsatisfactory purchases, while false negative interactions [12] can result from limited exposure or restricted browsing opportunities.

To tackle the challenge of noisy implicit feedback, denoising has become a significant focus in recommendation research, which can be broadly categorized into two main approaches:

- **Denoising based on side information.** Early studies [3, 10, 70] utilize user dwell time and gaze patterns to identify noise. Subsequent work incorporates sequence [66] and multi-behavior data [16, 58, 65] for more effective noise detection. Recent approaches integrate external knowledge graphs [22, 40, 72] or social graphs [9, 61] to better model user preferences. However, these methods can incur high data collection costs, and large-scale graphs may introduce additional noise, such as irrelevant attributes diluting user signals [12] or simplistic integrations amplifying noise [72].

- **Denoising driven by interaction data.** These methods utilize data selection and weighting strategies. Selection-based approaches [13, 29, 41] identify and filter noisy interactions by analyzing data features or employing decision networks. For example, [41] introduces an adaptive training strategy, while [13, 29] develop networks to exclude noisy samples. Reweighting-based methods [41, 45] adjust sample weights during training to mitigate noise effects, such as T-CE [41] which uses training loss for noise identification, BOD [45] which leverages interaction-derived priors with a bi-level optimization process.

Despite the effectiveness of interaction data-driven methods, they usually exhibit notable limitations. Firstly, they focus on learning user preferences from interaction data to identify noise. However, limited observational data result in only a partial understanding of user preferences, particularly in recognizing interactions that signal new interests or exploration tendencies [5, 35]. For instance,

in Figure 1(a), the pink area represents the user's true preference space $P$, while the green area denotes the observable preference space $\hat{P}$. The intersection $P \cap \hat{P}$ reflects the true preferences inferred from observational data. Interactions deemed as noise often consist of data inconsistent with currently learned preferences. For example, if an art enthusiast accidentally clicks on a gardening video, it may be labeled as noise, but it might indicate a latent interest in gardening sketches. Secondly, some studies rely on predefined assumptions [29, 44] in the noise identification process. For instance, [41] judging high-loss samples in training as noise, which inadequately captures user preferences and potential associations during noise identification (*e.g.,* links between fine arts and gardening). Consequently, it will diminish the model's effectiveness in denoising. To enhance the understanding of user preferences, large language models (LLMs) [1, 52, 69] present a promising direction due to their extensive world knowledge and reasoning capabilities. Recent studies [33, 49] have explored the application of LLMs in recommendation systems to improve the robustness of user representations by incorporating additional semantic and textual information. However, these approaches primarily enhance the semantic richness of representations while they insufficiently leveraging the potential of LLMs for denoising.

To explore the potential of LLMs for denoising in recommendation, we must address several significant challenges.

- **C1: How can LLMs effectively mine information relevant to denoising?** LLMs excel at processing textual information, allowing us to expand and enrich semantic insights that can inform denoising efforts. However, the interactive data represented in the graph structure of users and items contains rich collaborative information that is also valuable for denoising. Unfortunately, LLMs struggle to process this complex graph data effectively.
- **C2: How can we utilize the information generated by LLMs for denoising?** While LLMs can produce additional knowledge for denoising, they may also generate hallucinations [62], making direct application potentially suboptimal. Thus, it is crucial to consider how to constrain the knowledge generated by LLMs to align with the specific prediction targets in recommendations.

To address these challenges, we propose the **L**arge **La**nguage Model-enhanced **R**ecommendation **D**enoiser (LLaRD), a novel framework designed to develop recommendation models that are robust to noisy data. LLaRD consists of two main components: a knowledge generation module and a knowledge-enhanced denoising module. To tackle **C1**, the knowledge generation module leverages LLMs to extract two types of denoising-related knowledge: 1) **Preference knowledge**. Utilizing the inherent world knowledge of LLMs, we enrich the semantic information of the data through the analysis, reasoning, and refinement of text and interaction data. This process extrapolates the scope of observational data and infers user and item preferences more comprehensively. 2) **Relation knowledge**. We implement a novel chain-of-thought (CoT) prompting strategy [48, 56, 57] over graph structures to expand relation knowledge by iteratively reasoning about connections among users, items, and their neighborhood subgraphs. This approach encourages LLMs to consider key collaborative information hidden within the graph structure, thereby capturing relation knowledge pertinent to denoising. To address **C2**, the knowledge-enhanced denoising module is

built upon the Information Bottleneck (IB) [37, 47, 53]. It maximizes the mutual information across denoised data, generated knowledge, and recommendation targets, while minimizing the mutual information between the denoised data and the original data. This mechanism further filters out knowledge irrelevant to denoising from the information generated by LLMs, reducing the integration of irrelevant information, such as hallucinations, and thereby enhancing denoising performance. As illustrated in Figure 1(b), we anticipate that LLMs will improve the learning process of the denoising model, enabling it to more accurately capture the trajectory of true user preferences (orange arrow) and extensively encompass the preference area (pink region). In summary, our approach facilitates enhanced denoising by utilizing LLM-driven insights to improve recommendation performance.

The main contributions of this paper are summarized as follows:

- We identify and address the limitations of existing denoising recommendation methods, proposing a novel application of LLMs' world knowledge and reasoning capabilities to enhance the performance of recommendation models.
- We introduce LLaRD, a framework that integrates knowledge generation and knowledge-enhanced denoising strategies to leverage the capabilities of LLMs for achieving noise-robust recommendation models.
- We validate the effectiveness of LLaRD through extensive experiments on three benchmark datasets and two mainstream backbone models, demonstrating the framework's superior performance in denoising recommendation.

## 2 PRELIMINARIES

### 2.1 Denoising Recommendation

Let the user set be $\mathcal{U} = \{u\}$ and the item set be $\mathcal{I} = \{i\}$, with $|\mathcal{U}|$ and $|\mathcal{I}|$ representing the number of users and items, respectively. The interaction matrix is $\mathbf{R} \in \{0, 1\}^{|\mathcal{U}| \times |\mathcal{I}|}$, where $r_{ui} = 1$ indicates that user $u$ has interacted with item $i$. Given the interaction data $\mathcal{D} = \{(u, i, r_{ui}) | u \in \mathcal{U}, i \in \mathcal{I}\}$, we train a recommendation model $f$ with parameters $\theta_f$ to predict the likelihood of user interactions with unseen items, formulated as $\theta_f = \arg\min_{\theta_f} \mathcal{L}_{rec}(\mathcal{D})$, where $\mathcal{L}_{rec}$ is the recommendation loss. Using the BPR [34] loss as an example, we have:

$$\mathcal{L}_{rec} = \mathbb{E}_{(u,i,j) \sim \mathcal{D}} \log(\sigma(f(\mathbf{h}_u)^\top f(\mathbf{h}_i)) - f(\mathbf{h}_u)^\top f(\mathbf{h}_j)), \quad (1)$$

where $\mathbf{h}_{u/i} \in \mathbb{R}^d$ is the user/item representations, and $\sigma(\cdot)$ is the sigmoid function. The triple $(u, i, j)$ consists of user $u$, positive sample $i$, and negative sample $j$, sampled pairwise from $\mathcal{D}$. While $r_{ui} = 1$ typically indicates a positive preference, observed interactions (*e.g.,* views, clicks, and purchases) may introduce noise that does not accurately reflect true preferences. The denoising recommendation task aims to learn a clean interaction matrix $\mathbf{R}^* \in \{0, 1\}^{|\mathcal{U}| \times |\mathcal{I}|}$ representing users' genuine preferences or to derive noise-free representations $\mathbf{h}_{u/i}^* \in \mathbb{R}^d$ from the noisy data.

### 2.2 Information Bottleneck

The Information Bottleneck (IB) [37, 39, 53] is a powerful framework rooted in information theory, commonly used for representation learning. Its goal is to enhance the robustness of learned

representations for downstream tasks by discarding task-irrelevant information from the input data. We give the following definition:

**Definition 1 (Information Bottleneck).** *Let $X \in \mathcal{X}$ and $Y \in \mathcal{Y}$ be random variables with joint distribution $p(X, Y)$, where $X$ contains information relevant to $Y$. The relevant information is quantified by the mutual information $I(X; Y)$. The IB framework seeks the most informative yet compressed representation $Z$ by optimizing the objective: $\max_Z \{I(Y; Z), \ s.t. \ I(X; Z) \leq I_c\}$, where $I_c$ is the information constraint between $X$ and $Z$.*

By introducing a Lagrange multiplier $\lambda$, the constrained optimization is reformulated as an unconstrained objective: $\max_Z I(Y; Z) - \lambda I(X; Z)$. The IB principle is widely applied to generalization and denoising tasks. Several studies [37, 53] employ the Graph Information Bottleneck (GIB) principle to identify stable subgraphs to enhance model generalization, while methods like CGI [47] leverage the IB framework for denoising recommendation.

## 2.3 Chain-of-Thought Prompting

Chain-of-Thought (CoT) prompting [42, 48, 56, 57] enhances the reasoning capabilities of LLMs by guiding them to generate intermediate reasoning steps structured as $<$ input, thoughts, output $>$ instead of directly producing answers. This approach improves both interpretability and accuracy, particularly for tasks requiring multi-step reasoning or logical deductions.

**Definition 2 (CoT Prompting).** *CoT prompting directs a language model to produce a sequence of intermediate reasoning steps $R$ before generating the final output $Y$, given an input prompt $X$. Mathematically, this framework models the output $Y$ as:*

$$p(Y|X) = \sum_R p(Y|R, X), p(R|X). \tag{2}$$

*This decomposition transforms complex tasks into manageable subtasks, enhancing the reasoning capabilities of model.*

By generating structured reasoning steps, CoT prompting enables more accurate and reliable responses in complex tasks.

## 3 METHODOLOGY

In this section, we introduce the **L**arge **La**nguage Model-enhanced **R**ecommendation **D**enoiser (LLaRD). As illustrated in Figure 2, it comprises two knowledge generation modules and a denoising module. Below, we provide a detailed overview of each component.

## 3.1 Preference Knowledge Generation

In this module, we extract semantic preference information from textual data and user-item interactions despite inherent data noise. For example, the Amazon-Book dataset includes descriptions with irrelevant attributes, and reader reviews are often subjective and unstructured, featuring imaginative content, citations, or counterfactual statements. These factors complicate the direct extraction of meaningful preference semantics. To address this issue, we adopt methods from prior studies [33, 55], utilizing LLMs for text denoising and preference knowledge reasoning. We design system prompts $S_u$ and $S_i$ for users and items, respectively, and construct configuration texts $\mathcal{T}_u = \{T_u^1, T_u^2, ..., T_u^{|\mathcal{U}|}\}$ and $\mathcal{T}_i = $

$\{T_i^1, T_i^2, ..., T_i^{|\mathcal{I}|}\}$ for each user and item as follows:

$$T_u^k = \text{Item\_title} \parallel \text{Item\_description} \parallel \text{User\_comments}, \tag{3}$$

$$T_i^k = \text{Item\_title} \parallel \text{Item\_category} \parallel \text{Item\_description}. \tag{4}$$

The reasoning process of profile information is defined as:

$$\mathcal{P}_u, \mathcal{P}_i = \text{LLM}([S_u \| \mathcal{T}_u], [S_i \| \mathcal{T}_i]), \tag{5}$$

where $\text{LLM}(\cdot)$ denotes the LLM reasoning process, $\parallel$ denotes the concatenation of the system prompt and configuration texts. $\mathcal{P}_u$ and $\mathcal{P}_i$ denote the profile information for each user and item, respectively. While LLMs effectively refine user preferences and item features, integrating extensive textual knowledge for collaborative analysis across thousands of users and items leads to semantic imprecision and high token inference costs. To mitigate these issues, we propose a keyword condensation technique for each user and item, reducing semantic ambiguity and enabling incremental updates to preference semantics. This approach accommodates the dynamic nature of users and items, ensuring robust and efficient preference extraction. Furthermore, we enhance system prompts by introducing the $S'_{u/i}$ which guides LLMs to refine the keywords of user preferences and item features based on the obtained profile information $\mathcal{P}_u$ and $\mathcal{P}_i$. The keyword generation process is defined as follows:

$$\mathcal{A}_u, \mathcal{A}_i = \text{LLM}\left([S'_u \| \mathcal{P}_u], [S'_i \| \mathcal{P}_i]\right), \tag{6}$$

where $\mathcal{A}_u = \{A_u^1, A_u^2, ..., A_u^{|\mathcal{U}|}\}$ and $\mathcal{A}_i = \{A_i^1, A_i^2, ..., A_i^{|\mathcal{I}|}\}$. We then combine the profile information $\mathcal{P}_{u/i}$ with keywords $\mathcal{A}_{u/i}$ to form the preference knowledge $\mathcal{F}_{u/i}$. The preference knowledge $\mathcal{F}_{u/i}$ is converted into token sequences, resulting in token embedding matrices $\mathbf{T}_{u/i} = \{\mathbf{t}_1, \mathbf{t}_2, ...\}$. These token embeddings are processed through a multi-layer perceptron (MLP) network $W_t$ to generate semantic embeddings for each user and item:

$$\tilde{\mathbf{E}}_u, \tilde{\mathbf{E}}_i = W_t(\text{LLM}([\mathbf{T}_u, \mathbf{T}_i])) + b. \tag{7}$$

Finally, we encapsulate the obtained preference semantic embeddings $\tilde{\mathbf{E}}_u$ and $\tilde{\mathbf{E}}_i$ into the preference knowledge $\mathcal{K}_p$ as:

$$\mathcal{K}_p = \left\{\tilde{\mathbf{E}}_u = \{\tilde{\mathbf{e}}_{u1}, \tilde{\mathbf{e}}_{u2}, ...\}, \tilde{\mathbf{E}}_i = \{\tilde{\mathbf{e}}_{i1}, \tilde{\mathbf{e}}_{i2}, ...\}\right\}. \tag{8}$$

## 3.2 Relation Knowledge Generation

Previous studies utilizing LLMs to infer user preferences from interaction sequences often struggle to capture multi-hop relationships and long-path dependencies essential for understanding complex interactions. In contrast, our approach leverages the reasoning capabilities of LLMs over graph-structured data. By integrating preference semantics with collaborative information, we enable LLMs to identify associative semantics among multiple interaction nodes. Furthermore, we iteratively infer additional interaction edges to construct a relation knowledge graph, enhancing the graph learning process and improving the denoising of implicit feedback.

*3.2.1* ***User-Centric CoT Reasoning Framework.*** The collaborative information within the user-item interaction graph is invaluable for denoising. However, LLMs often struggle to achieve strong reasoning performance when dealing with complex interconnected data. To address this, we introduce a user-centric CoT reasoning framework. It meticulously designs inputs for multi-hop

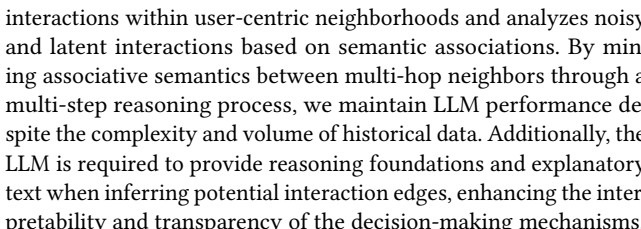

**Figure 2: The overview of the proposed LLaRD framework.**

interactions within user-centric neighborhoods and analyzes noisy and latent interactions based on semantic associations. By mining associative semantics between multi-hop neighbors through a multi-step reasoning process, we maintain LLM performance despite the complexity and volume of historical data. Additionally, the LLM is required to provide reasoning foundations and explanatory text when inferring potential interaction edges, enhancing the interpretability and transparency of the decision-making mechanisms.

**Step1: Preference Ratings.** We represent the preference of each user for items in their interaction sequence using a three-tier rating system: {*High, Medium, Low*}. For a user $u$, given the preference knowledge containing profile information and preference keywords, along with the interaction sequence $\mathcal{N}_u^{(1)} = \{i_1, i_2, ...\}$ and attribute keywords list $A_i^k$ for each item $i_k$, we follow the steps referring the Figure 2 for LLM inference. The output is a rated interaction sequence $\mathcal{N}_u^{Rated} = \{(i_1, l_{ui_1}), (i_2, l_{ui_2}), ...\}$, where $i_k$ denotes an item interacted with by user $u$, and $l_{ui_k} \in \{High, Medium, Low\}$ represents the user's preference rating for $i_k$.

**Step2: Noise Identification.** Building on Step 1, we enable the LLM to identify noise among interactions rated as *Low*, denoted by $\mathcal{N}_{u(low)}^{(1)} = \{i_k \in \mathcal{N}_u^{(1)} \mid l_{ui_k} = Low\}$. The set of noise interactions is defined as:

$$\mathcal{I}_u^{Noise} = \{i_k \in \mathcal{N}_{u(low)} \mid \text{LLM identifies } i_k \text{ as noise}\}. \quad (9)$$

Consequently, the noise interaction edges for each user are represented by:

$$\mathcal{E}^{Noise} = \{(u, i_k) \mid u \in \mathcal{U}, i_k \in \mathcal{I}_u^{Noise}\}. \quad (10)$$

By rigorously analyzing the semantic associations between user preferences and item attributes, our approach minimizes the misclassification of interactions that may reflect latent user interests. This sophisticated semantic analysis enables the model to discern and retain interactions that, although rated *Low*, may indicate emerging or subtle preferences.

**Step3: Collaborative Enhancement.** We perform second-hop neighbor exploration within the neighborhood of user $u$ to identify users with similar preferences, constructing enhanced collaborative interactions through semantic associations. Utilizing the preference ratings from Step 1, we focus on items rated as *High*, defined as $\mathcal{N}_{u(high)}^{(1)} = \{i_k \in \mathcal{N}_u^{(1)} \mid l_{ui_k} = High\}$. The set of second-hop neighbors is then determined by: $\mathcal{N}_u^{(2)} = \bigcup_{i_k \in \mathcal{N}_{u(High)}} U_{i_k} \setminus \{u\}$, where $U_{i_k}$ represents users who have interacted with item $i_k$, $\bigcup$ represents the union operation, and $\setminus \{u\}$ ensures that user $u$ is excluded from their own set of neighbors. Subsequently, we identify collaboratively enhanced users through LLM inference:

$$\mathcal{U}_u^{Collab} = \{u_k \in \mathcal{N}_u^{(2)} \mid \text{LLM identifies } u_k \text{ as enhancement}\}. \quad (11)$$

The corresponding set of collaborative enhancement interaction edges for each user is represented as:

$$\mathcal{E}^{Collab} = \{(u, u_k) \mid u \in \mathcal{U}, u_k \in \mathcal{U}_u^{Collab}\}. \quad (12)$$

This collaborative enhancement leverages semantic associations to connect users with similar high-preference interactions, thereby enriching the recommendation capability to accurately discern and predict user preferences.

**Step4: Interests Exploration.** In this step, we utilize LLM reasoning to explore interests within the third-hop neighborhood of user $u$. To prevent an exponential growth of high-order neighbors in the interaction graph, we selectively retain only interaction edges labeled as *High*, emphasizing their importance in accurately reflecting user preferences. Building on the preference intensities from previous steps and the analysis of first- and second-order neighbors, we infer potential interest interactions among third-order neighbors, defined as: $\mathcal{N}_u^{(3)} = \bigcup_{u_k \in \mathcal{N}_{u(High)}^{(2)}} I_{u_k} \setminus \mathcal{N}_u^{(1)}$. We then identify the set of interest items for user $u$ as:

$$\mathcal{I}_u^{Interests} = \{i_k \in \mathcal{N}_u^{(3)} \mid \text{LLM identifies } i_k \text{ as interests}\}. \quad (13)$$

The corresponding set of interest interaction edges is represented by:

$$\mathcal{E}^{Interests} = \{(u, i_k) \mid u \in \mathcal{U}, i_k \in \mathcal{I}_u^{Interests}\}. \tag{14}$$

Utilizing our user-centric CoT reasoning framework, we integrated collaborative information from the interaction graph with preference semantics. This integration enabled the identification of potential interactions that accurately reflect users' true preferences and encapsulate associative semantics. Through this multi-step reasoning process, we effectively capture the underlying association semantics, enhancing the ability of discerning and predicting nuanced user preferences and improving recommendation.

*3.2.2 Relation Knowledge Construction.* To effectively leverage the reasoning results, we construct the above three distinct groups of interaction edges as relation knowledge:

$$\mathcal{K}_r = \{\mathcal{E}^{Noise}, \mathcal{E}^{Collab}, \mathcal{E}^{Interests}\}. \tag{15}$$

Subsequently, we integrate this relation knowledge into the original interaction graph $\mathcal{G} = (\mathcal{U}, \mathcal{I}, \mathcal{E})$, where $\mathcal{U}$ and $\mathcal{I}$ represent the sets of users and items, respectively, and $\mathcal{E} = \{(u, i) \mid u \in \mathcal{U}, i \in \mathcal{I}, r_{ui} = 1\}$ denotes the existing interaction edges. The enriched interaction graph $\mathcal{G}_{rel}$ is formulated as:

$$\mathcal{G}_{rel} = \left(\mathcal{U}, \mathcal{I}, (\mathcal{E} \setminus \mathcal{E}^{Noise}) \cup \mathcal{E}^{Collab} \cup \mathcal{E}^{Interests}\right). \tag{16}$$

This enriched graph incorporates the relation knowledge by removing noise interactions and adding collaborative and interest-based interactions. This integration enhances the downstream denoising learning process, enabling more accurate and semantically rich preference extraction.

## 3.3 Knowledge-enhanced Denoising

After generating denoising knowledge, it is essential to use this to guide the denoising process. To achieve this, we propose a knowledge-enhanced denoising learning approach. As illustrated in the lower half of Figure 2, this approach includes a mask generator and a knowledge-guided information bottleneck framework.

*3.3.1 Mask Generator.* To effectively capture comprehensive user preferences and latent semantic associations within the graph structure, we incorporate additional injected knowledge. This enhanced understanding facilitates data selection, reweighting, and representation learning, enabling a robust recommendation model even when denoising is limited to observed data. We employ a mask generator to create a learnable mask that distinguishes noisy interaction edges from informative ones in the original interaction data. Specifically, given the interaction graph $\mathcal{G} = (\mathcal{V}, \mathcal{E})$, where $\mathcal{V} = \mathcal{U} \cup \mathcal{I}$ and $\mathcal{E} = \{(u, i) \mid u \in \mathcal{U}, i \in \mathcal{I}, r_{ui} = 1\}\}$, each edge is associated with a random variable $q \sim \text{Bernoulli}(\lambda)$. An edge is retained if $q = 1$ and deleted otherwise. We parameterize the Bernoulli parameter $\lambda$ using a MLP network $\Phi$ as $\lambda = \Phi(\mathbf{e}_u \| \mathbf{e}_i)$, where $\|$ denotes concatenation, and $\mathbf{e}_u, \mathbf{e}_i \in \mathbb{R}^d$ are the embeddings of user $u$ and item $i$ from the original interaction graph $\mathcal{G}$. To enable end-to-end training, we adopt the Gumbel-Softmax reparametrization trick, converting the discrete variable $q$ into a continuous variable in the range $[0, 1]$:

$$q = \sigma((\log \delta - \log(1 - \delta) + \lambda_m)/\tau), \tag{17}$$

where $\tau$ is the temperature hyperparameter and $\delta \sim \text{Uniform}(0, 1)$. As $\tau \to 0$, $q$ approaches a binary value. Finally, we obtain the masked graph $\mathcal{G}' = (\mathcal{U}, \mathcal{I}, \mathcal{E}')$, where $\mathcal{E}' = \{(u, i) \mid (u, i) \in \mathcal{E}, q_m \to 1\}$. This denoised graph retains only the informative interaction edges deemed relevant by the mask generator, thereby enhancing the downstream denoising learning process.

*3.3.2 Knowledge-guided Information Bottleneck for Denoising.* Building on the Information Bottleneck (IB) principle, we present an optimization framework for denoising interaction graphs. Our dual objectives are to maximize the retention of user preference information in the denoised graph and to minimize the mutual information between the denoised and original graphs. To comprehensively capture true user preferences, we integrate supervisory signals from interaction data with additional knowledge from LLMs, encompassing both explicit preferences and latent semantic associations. This combined approach effectively guides the denoising process. The optimization objective is formally expressed as:

$$\max_{\mathcal{G}'} I(\mathcal{R}; \mathcal{G}') + \alpha I(\mathcal{K}_p, \mathcal{K}_r; \mathcal{G}') - \beta I(\mathcal{G}'; \mathcal{G}), \tag{18}$$

where $I(\mathcal{R}; \mathcal{G}')$ denotes the mutual information between recommendation targets $\mathcal{R}$ and the denoised graph $\mathcal{G}'$. $I(\mathcal{K}_p, \mathcal{K}_r; \mathcal{G}')$ incorporates the mutual information between preference knowledge $\mathcal{K}_p$ relation knowledge $\mathcal{K}_r$, and the denoised graph $\mathcal{G}'$ integrating additional supervisory signals from LLMs. $I(\mathcal{G}'; \mathcal{G})$ denotes the mutual information between the denoised graph $\mathcal{G}'$ and the original graph $\mathcal{G}$ Here, $\alpha$ and $\beta$ are the hyperparameters that balance the influence of knowledge integration and noise reduction, respectively. Next, we detail the implementation of each term in Equation (18).

**Term1: Maximizing Mutual Information with Recommendation Information.** The first term aims to maximize information relevant to the recommendation task. We maximizes mutual information with the task-related information within $\mathcal{G}'$ by minimizing the BPR loss:

$$\mathcal{L}_{rec} = \sum_{(u,i,j) \in \mathcal{D}} - \log \sigma(y'_{ui} - y'_{uj}) \quad y'_{ui} = \mathbf{h}'^{\top}_u \mathbf{h}'_i, \tag{19}$$

where $\mathcal{D} = \{(u, i, j) \mid (u, i) \in \mathcal{D}^+, (u, j) \in \mathcal{D}^-\}$ is the training set, $\mathbf{h}_{u/i}$ and $\mathbf{h}'_{u/i}$ are the user and item representations after $L$ GNN layers on $\mathcal{G}'$. Minimizing $\mathcal{L}_{rec}$ effectively maximizes $I(\mathcal{R}; \mathcal{G}')$, ensuring that the denoised graph retains essential preference information from recommendation prediction.

**Term2: Preference & Relation Knowledge Integration.** The second term promotes retaining information in the denoised graph $\mathcal{G}'$ that integrate with both preference knowledge $\mathcal{K}_p$ and relation knowledge $\mathcal{K}_r$. Given the collaborative embeddings $\mathbf{e}_u$ and the preference knowledge embedding $\tilde{\mathbf{e}}_u$ and $\tilde{\mathbf{e}}_i$ from $\mathcal{K}_p$, our optimization objective uses the InfoNCE [15] loss to denote:

$$\mathcal{L}_{prf} = \sum_{v \in \mathcal{V}} - \log \frac{\exp(\text{sim}(\mathbf{h}'_v, \tilde{\mathbf{e}}_v)/\tau')}{\sum_{v' \in \mathcal{V}', v' \neq v} \exp(\text{sim}(\mathbf{h}'_v, \tilde{\mathbf{e}}_{v'})/\tau')}, \tag{20}$$

which $\text{sim}(\cdot)$ is the cosine similarity function, and $\tau'$ is the temperature parameter. $\mathbf{h}'_v$ is the final representation on $\mathcal{G}'$ after $L$ GNN layers, and $\tilde{\mathbf{e}}_v$ are embeddings derived from preference knowledge $\mathcal{K}_p$ Minimizing $\mathcal{L}_{prf}$ enhances the agreement between $\mathcal{G}'$ and

preference knowledge, capturing user preferences within semantic information. For the relation knowledge $\mathcal{K}_r$, we treat the relation knowledge graph $\mathcal{G}_{rel}$ with embeddings $\hat{E}_u = \{\hat{e}_{u1}, \hat{e}_{u2}, ...\}$ and $\hat{E}_i = \{\hat{e}_{i1}, \hat{e}_{i2}, ...\}$ as an augmented view of the interaction graph. After $L$ GNN layers on $\mathcal{G}_{rel}$, we obtain representation $\hat{h}_u$ and $\hat{h}_i$. The optimization objectives is defined as:

$$\mathcal{L}_{rel} = \sum_{v \in \mathcal{V}} -\log \frac{\exp(\text{sim}(h'_v, \hat{h}_v)/\tau')}{\sum_{v' \in \mathcal{V}', v' \neq v} \exp(\text{sim}(h'_v, \hat{h}_{v'})/\tau')}, \quad (21)$$

where $\tau'$ is the temperature parameter and $\text{sim}(\cdot)$ is the cosine similarity function. Minimizing $\mathcal{L}_{rel}$ integrates $\mathcal{G}'$ with relation knowledge $\mathcal{K}_r$, capturing latent semantic associations within the graph structure.

**Term3: Minimizing Mutual Information for Denoising.** The third term facilitates the compression of information in the original interaction graph, filtering out of redundant interactions. Directly minimizing mutual information between two high-dimensional graph representations is computationally intractable. To overcome this, we utilize the Hilbert-Schmidt Independence Criterion (HSIC) as an approximation for mutual information between $\mathcal{G}$ and $\mathcal{G}'$.

First, we select appropriate kernel functions $k(\cdot)$ and $m(\cdot)$ for $\mathcal{G}$ and $\mathcal{G}'$, respectively. For instance, Gaussian kernels are employed:

$$k(h_v, h_j) = \exp\left(-\frac{\|h_v - h_j\|^2}{2\sigma_k^2}\right), \; m(h'_v, h'_j) = \exp\left(-\frac{\|h'_v - h'_j\|^2}{2\sigma_m^2}\right), \quad (22)$$

where $\theta_k$ and $\theta_m$ are kernel bandwidth parameter, $h$ and $h'$ are the user/item representation of $\mathcal{G}$ and $\mathcal{G}'$, respectively. Using these kernel functions, we compute the kernel matrices $K$ and $M$ from the $\mathcal{G}$ and $\mathcal{G}'$:

$$K = [k(h_v, h_j)]_{n \times n}, \quad M = [m(h'_v, h'_j)]_{n \times n}, \quad (23)$$

where $n$ is the number of users/items in the graph and $v, j \in [0, n]$. To center the kernel matrices and remove the mean, we apply the centering matrix $H = I - \frac{1}{n}11^\top$, where $I_n$ is the $n \times n$ identity matrix and $1$ is an $n$-dimensional vector of ones. The centralized kernel matrices are $\tilde{K} = HKH$ and $\tilde{M} = HMH$. Using the centralized matrices $\tilde{K}$ and $\tilde{M}$, we compute HSIC as an approximation of mutual information between $\mathcal{G}$ and $\mathcal{G}'$:

$$\text{HSIC}(\mathcal{G}, \mathcal{G}') = \frac{1}{(n-1)^2}\text{trace}(\tilde{K}\tilde{M}). \quad (24)$$

The loss term for information compression using HSIC is defined as:

$$\mathcal{L}_{comp} = \text{HSIC}(\mathcal{G}, \mathcal{G}') = \frac{1}{(n-1)^2}\text{trace}(\tilde{K}\tilde{M}). \quad (25)$$

Minimizing HSIC effectively reduces the mutual information between the original graph $\mathcal{G}$ and the denoised graph $\mathcal{G}'$, ensuring that $\mathcal{G}'$ retains only the information necessary for the recommendation task, thereby achieving maximum compression and eliminating redundant interactions.

**Model Optimization.** The overall loss function combines the BPR loss, preference & relation knowledge and information compression loss, defined as:

$$\mathcal{L} = \mathcal{L}_{rec} + \alpha(\mathcal{L}_{prf} + \mathcal{L}_{rel}) + \beta\mathcal{L}_{comp}, \quad (26)$$

where $\alpha$ and $\beta$ are hyperparameters that balance the contribution of the preference alignment loss and the information compression loss, respectively.

## 4 EXPERIMENTS

To evaluate the effectiveness of LLaRD, we carry out a series of experiments to address the following **R**esearch **Q**uestions:

- **RQ1:** How does LLaRD perform compared to various state-of-the-art denoising models when applied to different backbones?
- **RQ2:** How can we verify the effectiveness of denoising knowledge mined by LLMs in denoising learning?
- **RQ3:** How effectively can LLaRD help the model acquire robust representations mitigate noise issues?
- **RQ4:** Is LLaRD effective in boosting the performance of cold-start users?

### 4.1 Experimental Settings

We conduct experiments on three benchmark datasets: Steam, Yelp, and Amazon-Book. We use two backbone models: GMF [26] and LightGCN [17]. Our baseline methods consist of instance-level denoising and representation-level denoising. The instance-level method include WBPR [11], T-CE [41], R-CE [41], DeCA [44], SGDL [12], BOD [45] and DCF [18]. The representation-level method include SGL [51], SimGCL [63] and RLMRec [33]. More details of the dataset and implementation are provided in Appendix B.

### 4.2 Performance Comparison (RQ1)

To evaluate the effectiveness and generalizability of our framework, we compared our proposed LLaRD method with existing denoising baselines across three datasets and two backbone models. The following observations summarize our findings:

- Our proposed LLaRD consistently outperforms mainstream denoising techniques across all three datasets and both backbone models. On average, LLaRD surpasses the second-best model, BOD, by approximately 6.92% when integrated with GMF, and by 11.79% with LightGCN. Although BOD employs a bi-level optimization strategy to extract prior knowledge, it lacks a comprehensive understanding of preferences and mining the relational semantics within interaction samples, resulting in inferior performance compared to our method.
- Against interaction data-driven methods such as T-CE, DeCA, DCF, and SGDL, which are constrained to identifying patterns within observed data and and rely on training loss for noise identification, LLaRD demonstrates a substantial performance improvement ranging from 46.1% to 68.53%. This significant enhancement is attributed to our utilization of LLMs to infer user preferences beyond the available interaction data and the application of CoT reasoning to progressively uncover complex semantic associations within the interaction graph, thereby eliminating dependence on predefined assumptions.
- LLaRD outperforms robust representation learning methods by approximately 34.34% to 49.31%. The LLM-enhanced method, RLMRec, also achieves a significant 14.93% improvement over traditional approaches like SGL and SimGCL by aligning user preferences across semantic and collaborative spaces, demonstrating the effectiveness of LLMs in providing task-relevant

**Table 1: Overall performance comparison of different baselines on the backbone models. Bold numbers indicate the best performance, and underlined numbers indicate the second-best performance. "R" and "N" stand for Recall and NDCG, respectively.**

| Dataset | | Amazon-Book | | | | Yelp | | | | Steam | | | |
|---|---|---|---|---|---|---|---|---|---|---|---|---|---|
| Backbone | Method | R@10 | N@10 | R@20 | N@20 | R@10 | N@10 | R@20 | N@20 | R@10 | N@10 | R@20 | N@20 |
| GMF | Normal | 0.0506 | 0.0399 | 0.0740 | 0.0463 | 0.0437 | 0.0352 | 0.0787 | 0.0465 | 0.0603 | 0.0512 | 0.0984 | 0.0599 |
| | WBPR | 0.0513 | 0.0404 | 0.0753 | 0.0477 | 0.0440 | 0.0359 | 0.0793 | 0.0569 | 0.0599 | 0.0505 | 0.0984 | 0.0597 |
| | R-CE | 0.0664 | 0.0508 | 0.0995 | 0.0615 | 0.0550 | 0.0464 | 0.0894 | 0.0578 | 0.0636 | 0.0536 | 0.1030 | 0.0665 |
| | T-CE | 0.0679 | 0.0533 | 0.1017 | 0.0641 | 0.0535 | 0.0453 | 0.0871 | 0.0565 | 0.0641 | 0.0536 | 0.1029 | 0.0663 |
| | DeCA | 0.0814 | 0.0619 | 0.1237 | 0.0710 | 0.0600 | 0.0515 | 0.0981 | 0.0619 | 0.0677 | 0.0555 | 0.1047 | 0.0676 |
| | SGDL | 0.0975 | 0.0741 | 0.1489 | 0.0902 | 0.0683 | 0.0560 | 0.1098 | 0.0696 | 0.0704 | 0.0582 | 0.1084 | 0.0699 |
| | RLMRec | 0.0968 | 0.0728 | 0.1483 | 0.0896 | 0.0662 | 0.0548 | 0.1092 | 0.0693 | 0.0810 | 0.0654 | 0.1283 | 0.0811 |
| | BOD | 0.1009 | 0.0779 | 0.1520 | 0.0944 | 0.0706 | 0.0574 | 0.1126 | 0.0712 | 0.0718 | 0.0596 | 0.1135 | 0.0744 |
| | LLaRD | **0.1083** | **0.0851** | **0.1619** | **0.1027** | **0.0708** | **0.0578** | **0.1135** | **0.0723** | **0.0819** | **0.0657** | **0.1291** | **0.0817** |
| LightGCN | Normal | 0.0670 | 0.0495 | 0.1010 | 0.0613 | 0.0539 | 0.0452 | 0.0871 | 0.0566 | 0.0731 | 0.0627 | 0.1170 | 0.0784 |
| | WBPR | 0.0674 | 0.0496 | 0.1016 | 0.0620 | 0.0539 | 0.0450 | 0.0877 | 0.0571 | 0.0735 | 0.0629 | 0.1165 | 0.0777 |
| | T-CE | 0.0693 | 0.0530 | 0.1079 | 0.0715 | 0.0585 | 0.0501 | 0.0906 | 0.0612 | 0.0736 | 0.0624 | 0.1133 | 0.0754 |
| | DCF | 0.0723 | 0.0557 | 0.1112 | 0.0743 | 0.0614 | 0.0524 | 0.0926 | 0.0627 | 0.0768 | 0.0672 | 0.1164 | 0.0771 |
| | DeCA | 0.0832 | 0.0611 | 0.1291 | 0.0799 | 0.0652 | 0.0576 | 0.1092 | 0.0689 | 0.0827 | 0.0711 | 0.1288 | 0.0882 |
| | SGL | 0.1018 | 0.0791 | 0.1498 | 0.0949 | 0.0718 | 0.0603 | 0.1171 | 0.0759 | 0.0795 | 0.0671 | 0.1254 | 0.0833 |
| | SimGCL | 0.1109 | 0.0873 | 0.1538 | 0.1013 | 0.0709 | 0.0599 | 0.1146 | 0.0748 | 0.0576 | 0.0471 | 0.0903 | 0.0587 |
| | SGDL | 0.1135 | 0.0872 | 0.1675 | 0.1054 | 0.0800 | 0.0661 | 0.1323 | 0.0841 | 0.0933 | 0.0769 | 0.1458 | 0.0755 |
| | RLMRec | 0.1034 | 0.0788 | 0.1600 | 0.0960 | 0.0794 | 0.0652 | 0.1275 | 0.0815 | 0.0926 | 0.0746 | 0.1452 | 0.0924 |
| | BOD | 0.1244 | 0.0985 | 0.1777 | 0.1131 | 0.0922 | 0.0739 | 0.1432 | 0.0884 | 0.1001 | 0.0802 | 0.1469 | 0.0891 |
| | LLaRD | **0.1408** | **0.1126** | **0.2028** | **0.1326** | **0.0975** | **0.0809** | **0.1574** | **0.1008** | **0.1054** | **0.0868** | **0.1631** | **0.1059** |

**Table 2: The impact of different components in LLaRD.**

| | Amazon-Book | | | | Steam | | | |
|---|---|---|---|---|---|---|---|---|
| Ablation | R@10 | N@10 | R@20 | N@20 | R@10 | N@10 | R@20 | N@20 |
| LLaRD | **0.1408** | **0.1126** | **0.2028** | **0.1326** | **0.1054** | **0.0868** | **0.1631** | **0.1059** |
| w/o $MI_{min}$ | 0.1259 | 0.1009 | 0.1868 | 0.1205 | 0.0977 | 0.0800 | 0.1525 | 0.0982 |
| w/o $MI_{max}$ | 0.1301 | 0.1039 | 0.1856 | 0.1215 | 0.0949 | 0.0774 | 0.1494 | 0.0957 |
| w/o PK | 0.1385 | 0.1090 | 0.1983 | 0.1292 | 0.1012 | 0.0837 | 0.1559 | 0.1017 |
| w/o RK | 0.1369 | 0.1075 | 0.1947 | 0.1244 | 0.1001 | 0.0819 | 0.1532 | 0.0904 |

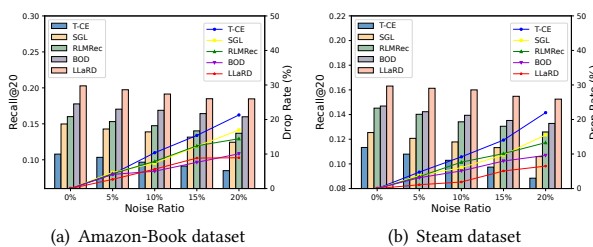

(a) Amazon-Book dataset  (b) Steam dataset

**Figure 3: Impact comparison *w.r.t.* noise ratio in added interaction data. The bars display Recall@20, while the curve shows the drop rate in performance.**

information. However, LLaRD surpasses these methods by not only ensuring robust representations but also addressing data-level denoising. It leverages higher-order associative semantics compared to RLMRec and enhances noise recognition capabilities, resulting in superior performance.

### 4.3 Ablation Study (RQ2)

To verify the effectiveness of denoising knowledge mined by LLMs and ensure its effective utilization in model learning, we conduct ablation studies to assess the contributions of various components within LLaRD. We design the following four model variants:

- w/o $MI_{min}$: Removes the process of minimizing mutual information between the denoised and original interaction graph.
- w/o $MI_{max}$: Removes the process of maximizing mutual information between the denoised graph and denoising knowledge.
- w/o PK: Removes the integration of preference knowledge in the denoising framework.
- w/o RK: Removes the integration of relation knowledge in the denoising framework.

As shown in Table 2, removing certain components leads to varying degrees of the performance degradation in LLaRD. The most significant decline occurs with the **w/o $MI_{min}$** variant, demonstrating

the effectiveness of our denoising approach based on the information bottleneck principle. Additionally, omitting either preference knowledge (**w/o PK**) or relation knowledge (**w/o RK**) results in performance reductions, highlighting their importance for denoising recommendations. Furthermore, when using the **w/o $MI_{max}$**, the performance decreases, underscoring the significance of denoising knowledge for model learning.

### 4.4 Model Benefits Analysis (RQ3 & RQ4)

**Robustness to Noisy Interactions.** To evaluate the robustness of LLaRD to noisy interactions, following previous studies [45, 51], we conducted experiments by introducing adversarial interaction examples (*i.e.,* 5%, 10%, 15%, and 20% negative user-item interactions) into the training set, while keeping the test set unchanged. Figures 3(a) and 3(b) present the results on the Amazon-Book and Steam datasets, respectively. This demonstrates that LLaRD consistently outperforms all baseline methods across all noise levels. Additionally, the performance drop of LLaRD remains relatively

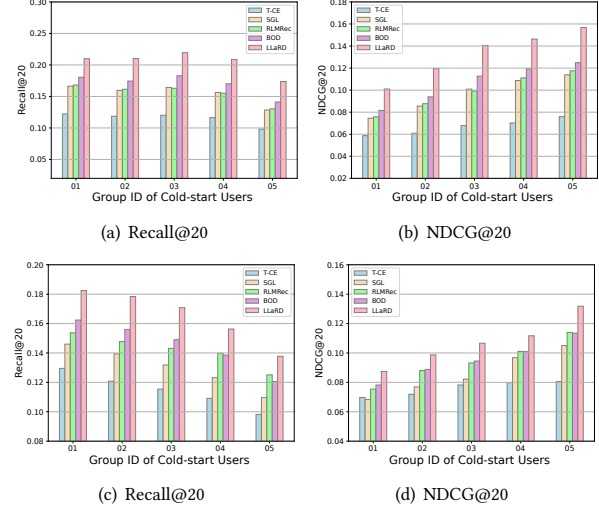

(a) Recall@20

(b) NDCG@20

(c) Recall@20

(d) NDCG@20

Figure 4: Recommendation performance over different cold-start user groups on Amazon-Book (upper) and Steam (lower) dataset.

stable compared to the baselines, highlighting its superior resilience to differing noise intensities. These results indicate that the denoising framework LLaRD effectively identifies and leverages useful patterns even in the presence of significant noise.

**Cold-Start Recommendation.** To evaluate the effectiveness in cold-start scenarios characterized by extremely sparse interaction data, we divide all users into five groups based on their interaction frequency. A lower group ID corresponds to sparser user activity and more severe cold-start issues. We compare LLaRD with various baseline methods across different cold-start levels. As shown in Figure 4, the results clearly indicate that LLaRD consistently outperforms baselines across all cold-start levels. This superior performance is attributed to the ability of LLaRD to derive preference knowledge and relation knowledge from LLMs, thereby enabling effective noise identification and robust modeling of both users and items, even in cold-start scenarios.

## 5 RELATED WORK

**Denoising in Recommendation.** Recommendation typically treat observed interactions as positive and unobserved ones as negative in implicit feedback [7, 12]. However, this approach can incorporate erroneous clicks or biased behaviors, leading to false positives and negatives that degrade user experience [38]. Existing denoising methods are generally categorized as follows: **1) Selection-Based Methods**: These methods [11, 50] filter out noisy feedback while retaining clean data. Early approaches [20, 31] use samplers based on data characteristics, whereas adaptive strategies later identify unreliable instances by detecting significant loss early in training. Recent techniques [29] employ deep reinforcement learning for effective noise removal. DCF [18] uses a dual-correction framework to identify noise through changes in sample loss over time. **2) Reweighting-Based Methods**: This approach assigns higher weights to informative interactions. Initial methods [44, 46] utilize training loss to assign lower weights to high-loss samples. Recent works

like DeCA [44] and BOD [45] have introduced novel evaluation criteria and optimization strategies for more accurate weight learning. **3) Side-Information-Based Methods and Special Strategies** [12, 51, 60]: Early approaches [3, 10, 70] utilize dwell time and annotations to detect noise. [16, 58, 66] incorporate sequential or multi-behavior data to capture unexpected interactions. [40, 72] have employed knowledge graphs to enhance preference modeling, facilitating denoising frameworks. In addition, there are some studies that learn robust representations by designing special denoising strategies. Early work [24, 36, 54] employ autoencoders to reduce noise in representations. [51, 63] leverage self-supervised learning on graph-structured data for greater stability. Despite their effectiveness, existing methods rely heavily on observed data and predefined assumptions to model user preferences and distinguish noise. In contrast, our approach leverages LLMs to acquire denoising knowledge, extracting inferred preference and relational semantics to capture noise interactions.

**LLMs in Recommendation.** LLMs [4, 68] have emerged as powerful tools for enhancing recommendation by leveraging deep semantic understanding and extensive pre-trained knowledge [6, 55, 67]. Some approaches [64, 64] capture latent preferences by generating textual tokens derived from user and item semantics, effectively modeling user preferences through LLMs' rich semantic capabilities. Other studies [14, 28, 30, 69] employ LLMs as recommenders by crafting specific instructions and fine-tuning them for recommendation tasks, utilizing their adaptability for tailored functionalities. Additionally, certain research [19, 25] adapts LLMs to downstream tasks using prompts without fine-tuning. For example, [19] introduce LLMs as zero-shot conversational recommender systems, while ToolRec [71] and RecMind [43] design CoT prompts to enable LLMs to handle complex reasoning within recommendation scenarios. Furthermore, methods [33, 49, 55, 59] generate rich-semantic embeddings and integrate reasoning knowledge into traditional models, improving understanding of user preferences and item features, thereby improving recommendation. Despite advancements in utilizing LLMs for various tasks, exploration in denoising recommendations remains limited. Our approach leverages LLMs to extract denoising-related knowledge, enhancing robustness by addressing noise interactions.

## 6 CONCLUSION

In this paper, we introduced LLaRD, a novel framework that leverages large language models (LLMs) to enhance the denoising process in recommendation. It improved denoising ability of the model by guiding LLMs to mine and inferred denoising-related knowledge from text and interaction data. Specifically, it first enriched semantic insights via LLMs, enabling a more comprehensive inference of user-item preferences. Then it employed a Chain-of-Thought (CoT) strategy over user-item interaction graphs to uncover relation knowledge relevant to denoising. Finally, the Information Bottleneck (IB) principle effectively aligned the denoised knowledge with recommendation targets. Through extensive empirical evaluations, we demonstrated that LLaRD significantly improves both denoising and recommendation accuracy compared to existing methods. Future work will explore further refinements to the framework and its applicability across diverse recommendation scenarios.

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

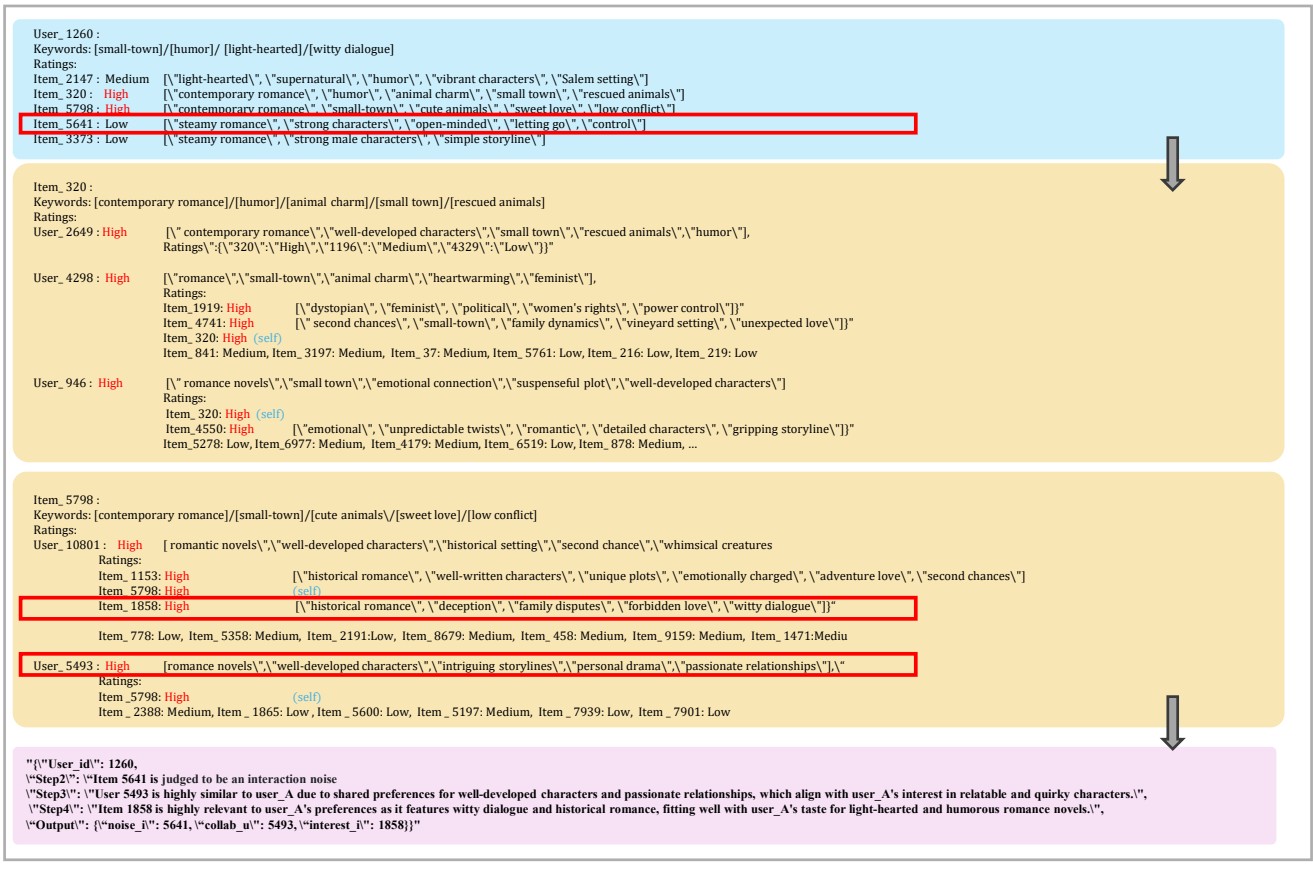

**Figure 5: The CoT reasoning case of LLaRD.**

## A  CASE STUDY

## B  MORE IMPLEMENTATION DETAILS

### B.1  Dataset Details

We conduct experiments on three benchmark datasets: Steam, Yelp, and Amazon-Book. Following the methods of [17, 33], we apply k-core filtering and divide each dataset into training, validation, and testing sets with a 3:1:1 ratio. Additionally, we remove interactions with ratings below 3, except for the Steam dataset, which does not include rating information and is therefore unfiltered. We provide the statistics of experimental datasets in Table 3

**Table 3: Statistics of experimental datasets.**

| Statistics | Amazon-Book | Steam | Yelp |
|---|---|---|---|
| # Users | 11,000 | 23,310 | 11,091 |
| # Items | 9,332 | 5,237 | 11,010 |
| # Interactions | 120,464 | 316,190 | 166,620 |
| # Density | 1.2e-3 | 2.6e-4 | 1.4e-4 |

### B.2  Evaluation Metrics

To ensure a fair evaluation and minimize bias, we adopt the all-rank protocol, considering all non-interacted items as candidates.

We assess performance using Recall@$N$ and NDCG@$N$, reporting average values for $N = 10$ and $N = 20$.

### B.3  Baselines and Backbone Models

We conduct experiments using two backbone models.

- **GMF** [26] decomposes the interaction matrix into implicit vectors and computes their element-wise product to capture features.
- **LightGCN** [17] is a widely adopted graph-based recommendation model. To demonstrate the effectiveness of our proposed method, we perform a fair comparison against traditional denoising techniques and state-of-the-art baselines.

Our baseline methods include instance-level denoising methods and representation-level denoising methods.

**Instance-level Denoising.**

- **WBPR** [11] is a sampling-based denoising method that assumes a negative item should be both highly popular and non-interacted.
- **T-CE** [41] is a re-weighting based method with truncated loss and dynamic thresholds during training.
- **R-CE** [41] is a re-weighting based method with reweighted loss and dynamic thresholds during training.
- **DeCA** [44] is a re-weighting based method addressing prediction disagreements of noisy interactions across models.

- **SGDL** [12] collects clean interactions at training onset, using similarity as a distinguishing criterion.
- **BOD** [45] models denoising as a bi-level optimization problem, extracting prior data information to generate weights.
- **DCF** [18] designs correction strategies for sample dropping and progressive labeling for precise denoising.

**Representation-level Denoising.**

- **SGL** [51] is a self-supervised framework performing graph contrastive learning with multiple views for robust representations.
- **SimGCL** [63] is a self-supervised framework adding uniform noise to embeddings to create contrasting views.
- **RLMRec** [33] utilizes LLMs to capture the complex user behavior semantics, enhancing recommendations through contrastive and generative techniques.

## B.4 Hyper-parameter Settings.

To ensure a fair comparison with the baselines, the dimension of representations and MLP is set to 64, and the GNN layer is set to 3, for all base models. The temperature value of contrastive learning from the range of 0.1, ..., 0.5. The temperature value of gumbel-max is 0.0001, and the hidden dim of attention is set to 64. During training, all methods are trained with a fixed batch size of 1024.We train all models using the learning rate 1e-3 with Adam optimizer without weight decay. We adopt the early stop technique based on the model's performance on the validation set. To generate the preference knowledge and relation knowledge, we leverage the Qwen model (specifically, qwen-long). For other parameters, we mainly use the official setting from the original paper and open-source code for fair comparisons. To allow for reproducibility, we also provide an anonymous code link of our work: https://anonymous.4open.science/r/LLaRD-5EE5.

