# OpenReview forum: "Unleashing the Power of Large Language Model for Denoising Recommendation"
_ACM.org/TheWebConf/2025/Conference — WWW 2025 Oral_

### Official Review · Reviewer_PxuU · 2024-11-21

**Novelty:** 5
**Technical Quality:** 5

**Review:**

Summary:
The paper introduces LLaRD, a novel framework that leverages the power of Large Language Models (LLMs) to enhance the denoising process in recommender systems. Recommender systems are crucial for personalizing user experiences, but they often rely on noisy and misleading implicit feedback data. Existing denoising methods typically involve external auxiliary information or data-driven strategies to reduce noise. However, these approaches face challenges due to the inherent limitations of the data and predefined assumptions, which hinder their ability to accurately identify noise. LLaRD aims to overcome these issues by harnessing the vast world knowledge and reasoning abilities of LLMs. By integrating LLMs into the denoising process, LLaRD significantly improves the quality of recommendations, potentially leading to more accurate and personalized user experiences.

Pros:
Novel Approach: The paper introduces an innovative approach by applying Large Language Models (LLMs) to denoise in recommender systems, which is a novel application. Additionally, the integration of Chain-of-Thought (CoT) prompting and the generation of two types of knowledge further enhance the originality of the approach.

High Quality of the Manuscript: The paper is well-structured, with clear explanations and strong visual aids (figures and diagrams). These illustrations effectively complement the text, making the methodology and proposed framework more accessible and easier to follow.

Cons:
Lack of Evaluation Models: The proposed method primarily relies on graph-based baseline models, but the paper does not provide sufficient analysis or experimental validation regarding its applicability to sequential recommendation models. Given the dynamic nature and temporal dependencies inherent in sequential data, it remains unclear whether the proposed approach would be equally effective in sequential recommendation tasks.

Time Complexity and Efficiency: While the paper introduces Chain-of-Thought (CoT) prompting and two types of generated knowledge to enhance the model's reasoning capabilities, it does not sufficiently analyze the time complexity and computational efficiency of the approach. The incorporation of CoT may introduce significant computational overhead. The lack of a detailed discussion on the associated time costs and efficiency trade-offs limits the practical applicability of the method.

**Questions:**

Could you clarify whether your approach can be effectively applied to sequential recommendation models? If so, how does the method perform in these contexts?


Could you provide an analysis of the time complexity of the proposed method?

**Reviewer Confidence:**

4: The reviewer is certain that the evaluation is correct and very familiar with the relevant literature

**Scope:**

4: The work is relevant to the Web and to the track, and is of broad interest to the community

---

### Official Review · Reviewer_dHZn · 2024-11-29

**Novelty:** 5
**Technical Quality:** 4

**Review:**

The manuscript presents a new framework, LLaRD, that leverages large language models (LLMs) to enhance the denoising process in recommender systems. The approach is well-structured and addresses a significant issue in the field of recommendation systems, particularly the challenge of noisy implicit feedback. The topic is interesting. However, the paper is not mature enough from the perspectives of contributions, novelty. I recommend authors to improve the paper. There are some problems needed to be solved.

1. A more detailed comparison with specific state-of-the-art methods would strengthen the argument and clarify how LLaRD uniquely addresses these challenges.

2. This paper lacks a comprehensive case study about denoising part.

3. Additional details on the experimental setup and validation processes would enhance the credibility of the findings.

**Questions:**

1. A more detailed comparison with specific state-of-the-art methods would strengthen the argument and clarify how LLaRD uniquely addresses these challenges.

2. This paper lacks a comprehensive case study about denoising part.

3. Additional details on the experimental setup and validation processes would enhance the credibility of the findings.

**Reviewer Confidence:**

3: The reviewer is confident but not certain that the evaluation is correct

**Scope:**

4: The work is relevant to the Web and to the track, and is of broad interest to the community

---

### Official Review · Reviewer_DSjK · 2024-11-29

**Novelty:** 5
**Technical Quality:** 3

**Review:**

This paper introduces the LLaRD framework, leveraging large language models (LLMs) and the Information Bottleneck principle to enhance denoising in recommendation systems.

Strengths
1. The proposed LLaRD framework, leveraging large language models (LLMs) for denoising, is novel and presents an interesting pipeline.
2. The paper is well-written and logically organized, making it easy to follow.
3. The motivation of the study is clearly articulated and compelling.

Weakness:
1. While the authors provide the code and datasets, I encountered an error due to a mismatch between the provided dataset and the code when attempting to deploy the example on my cluster. The issue could not be resolved by simply modifying the dataset name, preventing me from successfully replicating the results. This raises concerns regarding the reproducibility of the work.
2. In the related work section, the concept of using LLMs as denoisers for recommendation is partially discussed, but a relevant preprint ([1]) directly addressing a similar topic is missing. While I understand this preprint is not peer-reviewed, including a discussion of this work would better contextualize the novelty of LLaRD and highlight its unique contributions compared to [1].
3. Certain hyperparameters introduced in the proposed framework, such as α and β, which balance the different loss components, are not adequately discussed in the experimental section. Explaining their roles and sensitivity is essential for understanding the contributions of the individual loss terms and the overall loss balance.
4. The backbone models used in the experiments may not represent the current state-of-the-art in recommendation systems. Including at least one more recent backbone from the past two years would strengthen the evaluation and better reflect the framework’s performance against contemporary approaches.

[1] Wang, Bohao, et al. "Llm4dsr: Leveraing large language model for denoising sequential recommendation." arXiv preprint arXiv:2408.08208 (2024).

**Questions:**

Can this framework be integrated into an LLM-based recommendation system, such as LLaRA [2]?

[2] Liao, Jiayi, et al. "Llara: Large language-recommendation assistant." Proceedings of the 47th International ACM SIGIR Conference on Research and Development in Information Retrieval. 2024.

**Reviewer Confidence:**

3: The reviewer is confident but not certain that the evaluation is correct

**Scope:**

3: The work is somewhat relevant to the Web and to the track, and is of narrow interest to a sub-community

---

### Official Review · Reviewer_D55K · 2024-12-01

**Novelty:** 4
**Technical Quality:** 3

**Review:**

### Summary

The paper introduces LLaRD, a framework that leverages large language models (LLMs) for denoising recommendation systems, aiming to improve the accuracy of implicit feedback data. The proposed LLaRD framework shows substantial performance improvements over state-of-the-art approaches, supported by rigorous empirical evaluation. Despite the complexity and potential adoption challenges, the innovative methodology and practical relevance make this work a valuable contribution to the field.

### Pros
1. **Clear Structure**: The logical flow from introduction to methodology, experiments, and conclusions is easy to follow.
2. **Novel Integration**: The combination of LLMs, CoT reasoning, and the IB principle is a novel approach to enhancing recommendation systems.
3. **Empirical Validation**: The extensive experiments on multiple datasets demonstrate the effectiveness of the proposed framework.

### Cons
1. **Complexity**: The proposed method involves multiple sophisticated components, which may complicate implementation and integration into existing systems.
2. **Incremental Innovation**: While the integration is novel, the individual components (e.g., LLMs, CoT reasoning) are established in the literature.
3. **Dispersed Innovations**: There does not seem to be a strong correlation between the multiple innovation components
4. **Online Test**: This method lacks verification in actual scenarios.

**Questions:**

Please refer to the Cons listed above.

**Reviewer Confidence:**

3: The reviewer is confident but not certain that the evaluation is correct

**Scope:**

4: The work is relevant to the Web and to the track, and is of broad interest to the community

---

### Official Review · Reviewer_fUHs · 2024-12-07

**Novelty:** 6
**Technical Quality:** 6

**Review:**

### Summary

The authors propose a novel framework, LLaRD, to enhance denoising in recommendation systems by leveraging LLMs. LLaRD generates denoising-related knowledge using Chain-of-Thought reasoning and aligns it with recommendation targets through the Information Bottleneck principle. The authors demonstrate the effectiveness of their approach on three real-world datasets.

### Strengths

* The integration of LLMs and denoising in recommendation systems is a novel and promising direction.
* The authors have shared their code, promoting reproducibility and facilitating further research.
* The authors show that the proposed method consistently outperforms state-of-the-art baselines.

### Weaknesses

* The paper lacks clarity on how LLMs are used to generate keywords for users and items, which is a crucial component of the denoising process.
* The effectiveness of the method heavily relies on the quality of the LLM-generated knowledge and specific prompts. A potential limitation is the risk of overfitting to the single LLM or specific prompts. More analyses on this part are better.

**Questions:**

See review section.

**Reviewer Confidence:**

2: The reviewer is willing to defend the evaluation, but it is likely that the reviewer did not understand parts of the paper

**Scope:**

3: The work is somewhat relevant to the Web and to the track, and is of narrow interest to a sub-community